# KOBE: Cloud-native Open Benchmarking Engine for Federated Query Processors

Charalampos Kostopoulos, Giannis Mouchakis, Antonis Troumpoukis,
Nefeli Prokopaki-Kostopoulou, Angelos Charalambidis[0000−0001−7437−410X],
and Stasinos Konstantopoulos[0000−0002−2586−1726]

Institute and Informatics and Telecommunications,
NCSR "Demokritos", Greece
{b.kostopoulos,gmouchakis,antru,nefelipk,acharal,konstant}@iit.demokritos.gr

**Abstract.** In the SPARQL query processing community, as well as in the wider databases community, benchmark reproducibility is based on releasing datasets and query workloads. However, this paradigm breaks down for federated query processors, as these systems do not manage the data they serve to their clients but provide a data-integration abstraction over the actual query processors that are in direct contact with the data. As a consequence, benchmark results can be greatly affected by the performance and characteristics of the underlying data services. This is further aggravated when one considers benchmarking in more realistic conditions, where internet latency and throughput between the federator and the federated data sources is also a key factor. In this paper we present KOBE, a benchmarking system that leverages modern containerization and Cloud computing technologies in order to reproduce collections of data sources. In KOBE, data sources are formally described in more detail than what is conventionally provided, covering not only the data served but also the specific software that serves it and its configuration as well as the characteristics of the network that connects them. KOBE provides a specification formalism and a command-line interface that completely hides from the user the mechanics of provisioning and orchestrating the benchmarking process on Kubernetes-based infrastructures; and of simulating network latency. Finally, KOBE automates the process of collecting and comprehending logs, and extracting and visualizing evaluation metrics from these logs.

**Keywords:** Benchmarking, federated query processing, Cloud-native

## 1 Introduction

Data federation and distributed querying are key technologies for the efficient and scalable consuming of data in the decentralized and dynamic environment of the Semantic Web. Several federation systems have been proposed [10,4,2], each with their own characteristics, strengths, and limitations. Naturally, consistent and reproducible benchmarking is a key enabler of the relevant research,

as it allows these characteristics, strengths, and limitations to be studied and understood.

There are several benchmarks that aim to achieve this, but, similarly to the wider databases community, to release a benchmark amounts to releasing datasets, query workloads, and, at most, a benchmark-specific evaluation engine for executing the query load [5,9,8]. Research articles using these benchmarks need to specify what software has been used to implement the SPARQL endpoints, how it has been configured and distributed among hardware nodes, and the characteristics of these nodes and of the network that connects them to the federation system. Reproducing an experiment from such a description is a challenging and tedious task. Based on our own experience with federated query processing research we have been looking for ways to minimize the effort required and the uncertainty involved in replicating experimental setups from the federated querying literature. Our first step in that direction was to complement a benchmark we previously proposed [11] with Docker images of the populated triple store installations and of the federation systems used for that work.

In this paper we present KOBE,[1] an open-source[2] benchmarking engine that reads benchmark definitions and handles the distributed deployment of the data sources and the actual execution of the experiment. This includes instantiating a data source from dataset files, configuring and initializing the federation engine, connecting them into a virtual network with controlled characteristics, executing the experiment, and collecting the evaluation results. The main objective of KOBE is to provide a generic and controlled benchmarking framework where any combination of datasets, query loads, querying scenarios, and federation engines can be tested. To meet this goal, KOBE leverages modern Cloud-native technologies for the containerization and orchestration of different components.

In this paper we will first introduce the core concepts of a federated query processing experiment and the requirements for consistently and reproducibly carrying out such experiments (Section 2) and then present KOBE, its system components and how experiments are provisioned and orchestrated (Section 3). We then discuss how logs are collected and evaluation metrics visualized (Section 4), and how users can extend the library of benchmarks and federation engines to prepare their own experiments (Section 5). We close with a comparison to related systems (Section 6), conclusions and future work (Section 7).

## 2   Benchmarking Concepts and Requirements

We start by discussing the requirements for a benchmarking experiment of a federated query processor. First, we briefly introduce the main concepts of a federated query processing experiment:

**Data source:** An endpoint that processes queries. A data source is characterized by a dataset label, with data sources characterized by the same dataset serving the exact same data.

---

[1] Previously demonstrated in ISWC 2020, with extended abstract proceedings [6].
[2] See `https://github.com/semagrow/kobe`

**Benchmark:** A collection of data sources, the latency and throughput of these data sources, and a list of query strings. Benchmarks are defined independently of the federator that is being benchmarked.

**Federator:** A federated query processor that provides a single endpoint to achieve uniform and integrated access to the data sources.

**Experiment run:** A specific experiment, where (a) a specific federator has been configured to be able to connect to the data sources foreseen by the benchmark; and (b) the query load foreseen by the benchmark has been applied to the federator.

**Experiment:** The repetition of multiple runs of the same benchmark. An experiment is stateful, in the sense that the federator and data source instances are not terminated and maintain their caches and, in general, their state between runs.

Having these elements in place allows for the following tests, commonly used to evaluate query processing systems in general and federated query processing systems in particular:

– Comparing the first run for a query against subsequent runs; to understand the effect of caching.
– Observing if performance degrades for large numbers of runs by comparison to smaller numbers of runs; to understand if there are memory leaks and other instabilities.
– Observing if performance degrades for large numbers of experiments executed concurrently; to perform stress-testing.
– Comparing the performance of the same federation engine, on the same datasets, over different data sources; to understand the effect of current load, implicit response size limits, allocated memory, and other specifics of the query processing engines that implement the data sources.
– Comparing the performance of different federation engines on the same experiment; to evaluate federation engines.

Based on the above, we will now proceed to define the requirements for a benchmarking system that supports automating the benchmarking process.

## 2.1  Data Source Provisioning

In order to reliably reproduce evaluation results, there are several parameters of the data source implementation that need to be controlled as they affect evaluation metrics. These include the software used to implement the SPARQL endpoint and its configuration, the memory, processing power, disk speed of the server where it executes, the quality of the network connection between the data server and the federation engine, etc.

Replicating a specific software stack and its configuration can be captured by virtualization and containerization technologies, so we require that a benchmarking engine use recipes (such as a Dockerfile for Docker containers) that prepare each endpoint's execution environment.

The characteristics of the computing infrastructure where the data service executes and of the network connection between the data service and the federation engine can be naturally aggregated as the latency and throughput at which the federation engine receives data from it. So, one requirement from benchmarking engines is that latency and throughput can be throttled to a maximum, although other conditions might make a data service even less responsive than these maxima: e.g., a data source might be processing an extremely demanding query or might be serving many clients in a stress test scenario.

Based on this observation, we require that benchmarking engines allow the experiment description to include the latency and throughput between the data sources and the federation. And, in fact, that these parameters are specific to each data source. Technically, this requires that the architecture foresees a configurable proxy between the federator and each data source, so that each experiment can set this parameter to simulate the real behaviour of SPARQL query processors.

Naturally, this is in addition to the obvious requirement to control the data served and the way that data is distributed between data services.

## 2.2 Sequential and Concurrent Application of Query Workload

The benchmarking engine should automate the process of applying a query load to the federation engine. The queries that make up the query load should be applied either sequentially to evaluate performance on different queries or concurrently to stress-test the system.

Technically, a benchmarking system should include an orchestrator that can read such operational parameters from the experiment definition and apply them when serving as a client application for the federation engine.

## 2.3 Logs Collection and Analysis

One important requirement of a benchmarking system is that the experimenter can have easy access on several statistics and *key performance indicators* of each conducted experiment. An effective presentation of such indicators can offer to the experimenter the ability to compare the performance of different setups of the same benchmark (e.g., different federators or data sources) and to draw conclusions for a specific setup by examining time measurements for each phase of the query processing and several other metrics.

Metrics that are important for the experimenter to analyze the effectiveness of a federator in a specific benchmark, include the following:

– The *number of returned results* can be used to validate the correctness of the query processing by verifying that the federator returns the expected number of results. Naturally, this validation is incomplete as the results might have the correct cardinality and still be different from the correct ones. However, many errors can be very efficiently caught by simply comparing cardinalities before proceeding to the detailed comparison.

- The *total time to receive the complete result set* indicates how the engine performs overall from the perspective of the client. This is the most common key indicator that most benchmarks consider.
- Although different federated query processing architectures have been proposed, there is some convergence on *source selection*, *query planning*, and *query execution* as beeing the main query processing phases. Regardless of whether these phases execute sequentially or are adaptive and their execution is interwined, the *breakdown of the query processing time into phases* provides the experimenter with insights regarding the efficiency of the federation engine and how it can be improved.
- The *number of sources accessed* during processing a specific query can be used to evaluate the effectiveness of source selection in terms of excluding redundant sources from the execution plan.

The aforementioned key performance indicators can be computed by different pieces of software during an experiment execution. For instance, the first two metrics of the above list should be computed by the evaluator (i.e., the software that poses the queries to the federator), while the last two metrics can be computed only by the federation engine itself. In order for these metrics to be available to the experimenter, the benchmarking system must collect and process the log lines emitted by the federation engine and the other components. This will produce an additional requirement on the compatible format of the log lines of the systems under test.

## 3 The KOBE System

The *KOBE Benchmarking Engine (KOBE)* is a system that aims to provide an extensible platform to facilitate benchmarking on federated query processing. It was designed with the following objectives in mind:

1. to ease the deployment of complex benchmarking experiments by automating the tedious tasks of initialization and execution;
2. to allow for benchmark and experiment specifications to be reproduced in different environments and be able to produce comparable and reliable results;
3. to provide to the experimenter the reporting that is identified by the requirements in Section 2.

In the following sections we will present the architecture and components of KOBE and its key features.

### 3.1 Deployment Automation

One of the major tasks that KOBE undertakes is the deployment, distribution and resource allocation of the various systems (i.e., the database systems, the federator and others) that participate on a specific experiment. In order to achieve

this task, KOBE employs Cloud-native technologies to facilitate the deployment on cloud infrastructures. Each system is deployed in an isolated environment with user-defined computational resources and network bandwidth. In particular, KOBE leverages containerization technologies to support the deployment of systems with different environments and installation requirements. An immediate consequence of employing those technologies is that KOBE is open and can be extended with arbitrary federators and database systems.

KOBE consists of three main subsystems that control three aspects of the benchmarking process:

- The *deployment subsystem* that is responsible for deploying and initializing the components required by an experiment. This subsystem handles the allocation of computational resources for each component.
- The *networking subsystem* that is responsible for connecting the different components of an experiment and imposes the throughput and latency limitations described by the benchmark.
- The *logging subsystem* that manages the logs produced by the several components (i.e, the data sources, federators and evaluators) and produces meaningful diagrams and graphs about the benchmarking process.

KOBE relies on Kubernetes[3] to allocate cluster resources for the benchmark execution. It deploys ephemeral containers with the individual components of a benchmarking experiment. The orchestration of that deployment and the communication with the underlying Kubernetes cluster is performed by the *KOBE operator*. The KOBE operator runs as a daemon and continuously monitors the progress of each running experiment in the cluster. This controller is also responsible for the interpretation of the experiment specifications (see Subsection 3.2) to complete deployment commands of the components of the experiment.

The network subsystem is controlled by Istio[4], a Cloud-native controller that tightly integrates with Kubernetes to provide a service mesh layer. The KOBE operator utilizes the functionality of Istio to setup the network connections between the data sources and the federating engine. The quality of those network connections can be controlled by the KOBE operator to provide the simulated behavior specified by the specific experiment. It is worth noting that those network links are established in the service mesh layer of the cluster and as a result one can have multiple experiments with different networking topologies running at the same time in the cluster.

The logging subsystem of KOBE is implemented as an EFK stack, a popular solution for a centralized, cluster-level logging environment in a Kubernetes cluster. EFK stack consists of (a) Elasticsearch[5], an object store where all logs are stored in a structured form, used for log searching, (b) Fluentd[6], a data collector which gathers logs from all containers in the cluster and feeds them into

---

[3] cf. `https://kubernetes.io`

[4] cf. `https://istio.io`

[5] cf. `https://www.elastic.co/elasticsearch`

[6] cf. `https://www.fluentd.org`

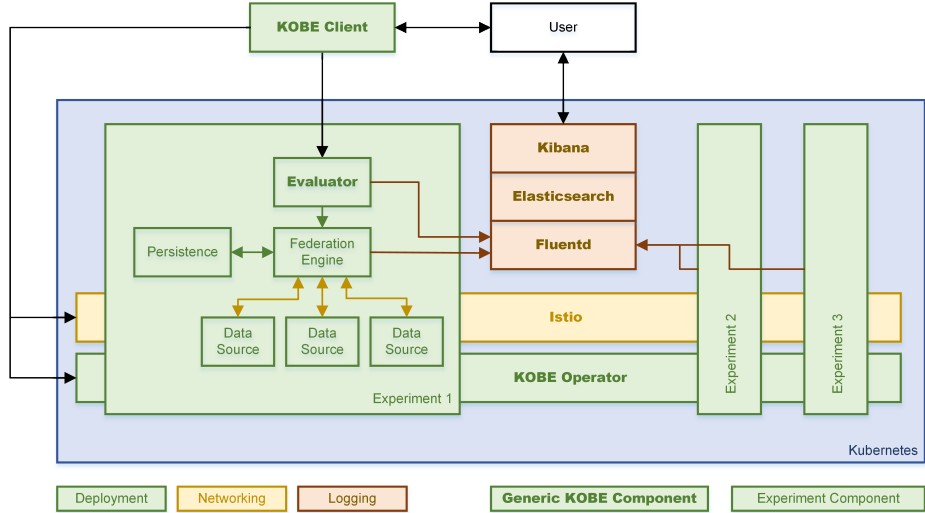

**Fig. 1.** Information flow through a KOBE deployment: The user edits configuration files and uses kobectl (the KOBE command-line client) to deploy and execute the benchmarking experiments, at a level that abstracts away from Kubernetes specifics. Experimental results are automatically collected and visualized using the EFK stack.

Elasticsearch, and (c) Kibana[7], a web UI for Elasticsearch, used for log visualization. Since the metrics of our interest are produced from the federator and the evaluator, and, as we will see in Section 4, these logs are of a specific form, Fluentd is configured to parse and to keep only the logs of these containers using a set of regular expression patterns for each type of KOBE-specific logs.

Figure 1 illustrates the relationships between the individual components and the information flow through this architecture. In a typical workflow, the user uses kobectl (the KOBE command-line client) to send commands to the KOBE operator. The operator, itself deployed as a container in the Kubernetes cluster, communicates with the Kubernetes API and with Istio in order to deploy the corresponding containers and establish the network between them. Moreover, a Fluentd logging agent is attached to each related container in order to collect the respective log output. The user also uses kobectl to provide a query load to the evaluator. The query evaluator is also deployed as a containerized application and is responsible for applying the query load to the federator and for measuring the latter's response.

During the execution of the experiment, Fluentd collects the log output from the evaluator, and parses it to extract evaluation metrics which are stored in Elasticsearch. If the federation engine is KOBE-aware, then it also produces log lines following the syntax understood by Fluentd so that fine-grained metrics about the different stages of the overall query processing are also computed and

---

[7] cf. `https://www.elastic.co/kibana`

stored in Elasticsearch. The user connects to Kibana to see visualizations of these metrics, where we have prepared a variety of panels specifically relevant to benchmarking federated query processors.

## 3.2 Benchmark and Experiment Specifications

An important aspect of benchmarking is the ability to reproduce the experimental results of a benchmark. KOBE tackles this important issue by defining declarative specifications of the benchmarks and the experiments. Those descriptions can be serialized in a human-readable format (we use YAML as the markup language) and shared and distributed as artifacts.

These specifications are grouped around the various components of an experiment including the benchmark, the evaluator, the data source systems, the data federator and the network topology. Typically, those specifications are partitioned in a series of files; each file includes informations about different elements of the experiment. For example, one specification describes a specific federator and a different specification includes information about the set of datasets and querysets.

The main idea of this organization is that each specification can be provided by a different role. For example, the federator (resp. dataset server) specification should be provided by the *implementor of the federator* (resp. *dataset server*). These specifications include, for example, details about the correct initialization of a federation engine. Moreover, the benchmark specification should be provided by the *benchmark designer* and the more specific details such as the computational resources and the network topology by the *experimenter*. The relevant pages of the online KOBE manual[8] give details about these parameters.

It is worth noting that the specifications are declarative in the sense that they describe the desired outcome rather than the actual steps one needs to follow to reproduce the experiment. The KOBE operator interprets these specifications as the necessary interactions with Kubernetes and Istio to deploy an experiment.

## 3.3 Experiment Orchestration

The KOBE operator is continuously monitoring for new experiment specifications that are submitted to KOBE by the user via a command-line client application. Upon a new experiment submission, the KOBE operator compiles new deployments for the data sources. The data sources consists of a list of dataset files, that is the serializable content of the dataset, and specifications about the database system that will serve this dataset. The deployment of a data source is performed in two phases: in the first phase the data files are downloaded and imported into the database system and in the second phase the system is configured and started for serving.

When all data sources are ready for serving, the federating engine is started. Similarly, the federating engine is deployed in two phases. In the first phase,

---

[8] `https://semagrow.github.io/kobe/references/api`

the federation of the specific instances of data sources is established. This includes the specific initialization process that a federation engine might need. For example, some engines need the generation of a set of metadata that depend on the specific datasets that they federate. The second phase start the actual federation service. After that, the network connections are established and the network quality characteristics are configured.

In that stage the experiment is ready to proceed with querying the federation. This is accomplished by an evaluator component that reads the query set from the benchmark specification and starts sending the queries to the endpoint of the federator. The evaluator is just another container that is deployed in the cluster. During the query evaluation, potential logs that are produced by the federation engine and the evaluator are collected and visualized to the user. The experiment completes when the evaluator finished with all the queries.

## 4 Collecting and Analysing Evaluation Metrics

In Section 2.3 we stipulated that benchmarking engines should include a mechanism that collects and analyzes the logs from multiple containers in order to compute evaluation metrics, and to present them to the experimenter in an intuitive way.

### 4.1 Collecting the Evaluation Metrics

In KOBE, the following benchmarking metrics are treated: the duration of the query processing phases (source selection, planning, and execution); the number of sources accessed during a query evaluation from the federator; the total time to receive the complete result set of a query; and the number of the returned results of a query. We assume that the federator and the evaluator calculate these metrics and produce a corresponding log message for each metric.

Notice, though, that many executions of several experiments can result in multiple query evaluations. As a result, many log messages that contain the same metric can appear. In order to differentiate between these query evaluations and to collect all logs that refer to the same query that belongs to a specific run of an experiment, each log message should also provide the following information:

**Experiment name:** This information is used to identify in which experiment the given query evaluation belongs.
**Start time of the experiment:** Since one experiment can be executed several times, this information is used to link to the given query evaluation with a specific experiment execution.
**Query name:** Each query has a unique identification name in an experiment. This information is used to refer to the name of the query in the experiment.
**Run:** Each experiment has several runs, meaning that the evaluation of a query happens multiple times in a specific experiment execution. This information identifies in which run of the experiment the given query evaluation belongs.

An important problem that arises is that this information is only available to the evaluator and cannot be accessed by the federator directly. Any heuristic workarounds that try to connect the evaluator log to the federator log using, for instance, the query strings would not work, as query strings are not unique. Especially in stress-testing scenarios, the exact same query string might be simultaneously executed multiple times, so that a combination of query strings and timestamps would not be guaranteed to work either. To work around this problem, the KOBE evaluator uses SPARQL comments to pass the query *experiment* id to the federator, and the latter includes those in its logs. Then, the federator can retrieve this information by parsing this comment. This approach has the advantage that even if a federation engine has not been modified to produce log lines that provide this information, the query string is still in a valid, standard syntax and the comment is ignored. The fine-grained time to complete each step in the typical federated query processing pipeline cannot be retrieved, but the experiment can proceed with the end-to-end query processing measurements provided by the evaluator.

## 4.2 Visualizing the Evaluation Metrics

In this subsection, we describe the visualization component of KOBE. In particular, we present the three available dashboards. For every dashboard we provide some screenshots of the graphs produced for some experiment runs.

**Details of a specific experiment execution** The dashboard of Figure 2 focuses on a specific experiment execution. It comprises:

1. Time of each phase of the query processing for each query of the experiment.
2. Total time to receive the complete result set for each query of the experiment.
3. Number of sources accessed for each query of the experiment.
4. Number of returned results for each query of the experiment.

The first and the third visualizations are obtained from the logs of the federator engine, if available. The second and the fourth visualizations are obtained from the logs of the evaluator, so they are available even for federators that do not provide KOBE-specific logs. The values in each visualization can be also exported in a CSV file for further processing.

As an example, we consider an experiment execution for the life-science (ls) query set of the FedBench benchmark for a development version of the Semagrow federation engine. This visualization can help us, for instance, to observe that the query execution phase of the federation engine dominates the overall query processing time in all queries of the benchmark except ls4.

**Comparisons of experiment runs** The dashboards depicted in Figure 3 and Figure 4 can be used to draw comparisons between several runs in order to directly compare different configurations of a benchmark. The dashboard of Figure 3 can be used for comparing several experiment executions. It consists of two visualizations:

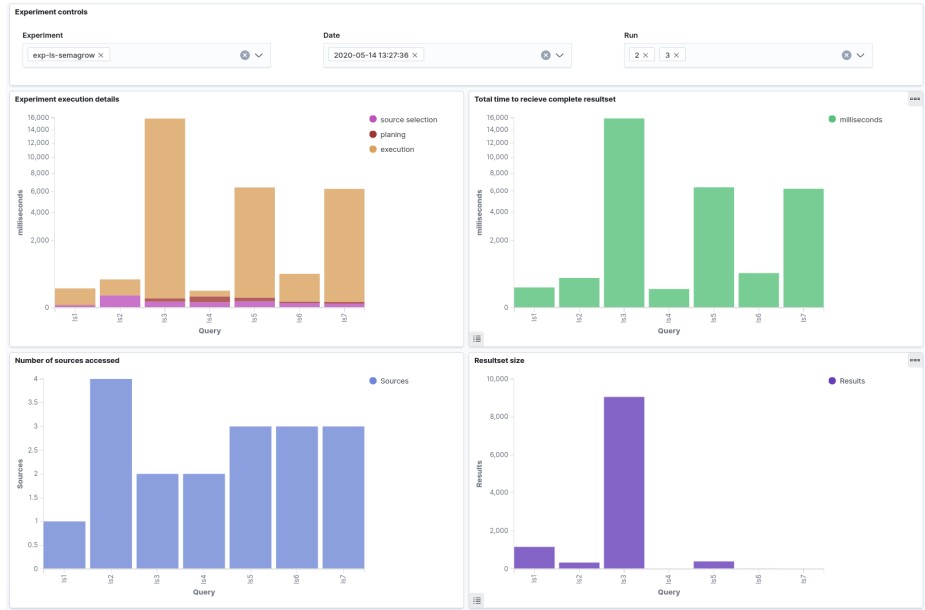

**Fig. 2.** Details of a specific experiment execution

1. Total time to receive the complete result set for each experiment execution.
2. Number of returned results for each specified experiment execution.

These visualizations are obtained from the logs of the evaluator. Each bar refers to a single query of the experiments presented. The dashboard of Figure 4 displays the same metrics. The main difference is that it focuses on a specific query and compare all runs of this query for several experiment executions. Contrary to the visualizations of the other two dashboards, each bar refers to a single experiment run, and all runs are grouped according to the experiment execution they belong to.

Continuing the previous example, we consider three experiment executions that refer to for the life-science queryset of FedBench; one for the FedX federator and two for the Semagrow federator. In Figure 3 we can observe that all executions return the same number of results for each query, and that the processing times are similar, with the exception of the ls6 query for the FedX experiment. Moreover, we can observe that all runs return same number of results, and that the processing times for each run are similar; therefore any caching used by the federators does not play any significant role in speeding up this query.

## 5 KOBE Extensibility

It is apparent that a well-designed and well-executed benchmarking experiment needs contributions from different actors. For example, a benchmark designer

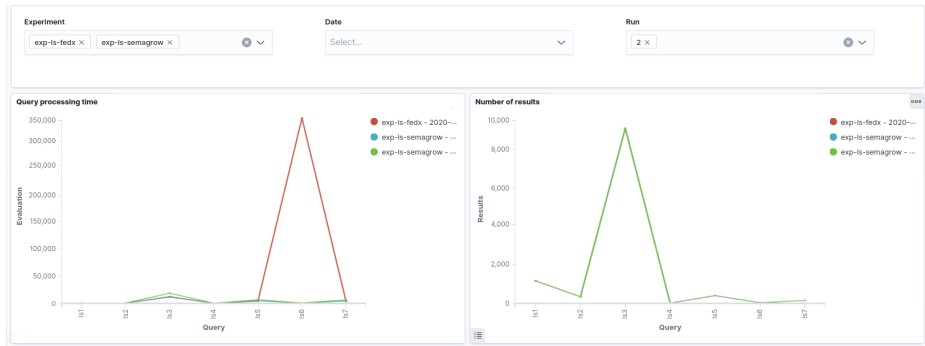

**Fig. 3.** Comparison of three experiment executions

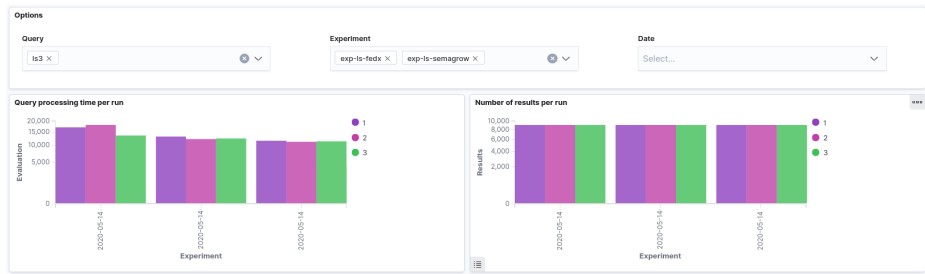

**Fig. 4.** Comparison of all runs of the ls3 query for three experiment executions

may provide a benchmark that is designed to compare a particular aspect of different federators. On the other hand, the specifications of each federator should ideally be provided by their respective implementors.

KOBE provides various extensibility opportunities and by design welcomes contributions from the community. In particular, KOBE can be extended with respect to the database systems, federators, query evaluators and benchmarks that comprise an experiment.

We currently provide specifications for two database systems, namely for Virtuoso[9] and Strabon[10] and for two federators, FedX [10] and Semagrow [2]. These systems have very different requirements in terms of deployment, providing strong evidence that extending the list of supported RDF stores will be straightforwrd.

We also provide a range of benchmark and experiment specifications for existing federated SPARQL benchmarks. Currently, the benchmarks that are already bundled with KOBE include the most widely used LUBM [5] and FedBench [9] benchmark. Moreover, we also include big RDF data benchmarks BigRDFBench

---

[9] cf. `https://virtuoso.openlinksw.com`

[10] cf. `http://strabon.di.uoa.gr`

[8] and OPFBench [11] and geospatial benchmarks GeoFedBench [12] and Geographica [3].

In the following, we briefly discuss the process of defining these specifications and give links to the more detailed walk-throughs provided in the online KOBE documentation.

### 5.1 Benchmarks and Experiments

Benchmarks are defined independently of the federator and comprise a set of datasets and a list of queries. Datasets are described in terms of the data and the system that should serve them. Data can be provided as a data dump to be imported in the database systems. For example, RDF data can be redistributed in the N-Triples format. Each dataset is characterized by its name and is parameterized by the URL where the data dump can be accessed. Queries of the benchmark are typically described as strings and annotated with the query language in which they are expressed; supporting heterogeneous benchmarks where not all data is served by SPARQL endpoints. A benchmark specification can also include network parameters, such as a fixed delay, or a percentage on which delay will be introduced as part of fault injection. The online KOBE manual provides walk-throughs for defining a new benchmark[11] and for tuning network parameters.[12]

An experiment that evaluates the performance of a federator over a given benchmark is defined using a strategy for applying the query load to the federator and the number of runs for each query of the experiment. The experimenter specifies an experiment by providing a new unique name for the experiment, the unique name of the benchmark and the federator specification. Moreover, an experiment includes a specific query evaluator, and the number of runs of the experiment. The query evaluator applies the query load to the federator. The one currently bundled with KOBE performs sequential querying, meaning that the queries of the benchmark are evaluated in a sequential manner. The online KOBE manual provides walk-throughs for defining a new experiment[13] and for extending KOBE with a new evaluator.[14] Furthermore, the manual also provides a walk-through for defining and visualizing new metrics.[15]

### 5.2 Dataset Servers and Federators

Dataset servers can be also integrated in KOBE. The dataset server specification contains a set of initialization scripts and a Docker image for the actual dataset server. The initialization scripts are also wrapped on isolated Docker containers and are used for properly initializing the database system. Typically, it includes

---

[11] `https://semagrow.github.io/kobe/use/create_benchmark`

[12] `https://semagrow.github.io/kobe/use/tune_network`

[13] `https://semagrow.github.io/kobe/use/create_experiment`

[14] `https://semagrow.github.io/kobe/extend/add_evaluator`

[15] `https://semagrow.github.io/kobe/extend/add_metrics`

the import of the data dump and indexing of the database. The dataset server specification may also include other parameters for network connectivity such as the port and the path to the listening SPARQL endpoint. A walk-through for adding a new dataset server is provided in the online KOBE manual.[16]

Federators can also be added to the KOBE system by providing the appropriate specification. That specification resembles the specification of a ordinary dataset server. The main difference is on the initialization phase of a federator. Typically, the initialization of a federator may involve the creation of histograms from the underlying datasets. Thus, in KOBE, the federator initialization is performed in two steps: the first step extracts needed information from each dataset and the second step consolidates that information and properly initializes the federator. As in the dataset server, the initialization processes are provided as containerized Docker images by the implementor of the federator. A walk-through for adding a federator is provided in the online KOBE manual.[17]

Federator implementors should also consider a tighter integration in order to benefit from the detailed log collection features for reporting measurements that can only be extracted by collecting information internal to the federator (Section 4). Therefore, a log line from a federator should be enhanced to include the evaluation metrics and the query parameters discussed in Section 2.3. More details about how a federator should be extended to provide detailed logs are given in the online KOBE manual.[18] This tighter integration is not a requirement, in the sense that the overall end-to-end time to evaluate the query and the number of returned results are provided without modifying the source code of the federation engine (as we have done in the case of FedX).

## 6 Comparison to Related Systems

To the best of our knowledge, the only benchmark orchestrator that directly targets federated query processors is the orchestrator distributed with the FedBench suite [9]. As also stated in the introduction, it is in fact the limitations of the FedBench orchestrator that originally motivated the work described here. Specifically, FedBench does not support the user with either container-based deployment or collecting federator logs to compute detailed metrics.

HOBBIT [7], on the other hand, is a Docker-based system aiming at benchmarking the complete lifecycle of Linked Data generation and consumption. Although HOBBIT tooling can support with collecting logs and visualizing metrics, HOBBIT as a whole is not directly comparable to KOBE. In the HOBBIT architecture, the benchmarked system is perceived as an opaque container that the system tasks and measures. KOBE exploits the premise that the benchmarked system comprises multiple containers one of which (the federator) is tasked and

---

[16] https://semagrow.github.io/kobe/extend/add_dataset_server

[17] https://semagrow.github.io/kobe/extend/add_federator

[18] Specifically, see the first step of the walk-through for adding a new federator. See also details about collecting logs to compute evaluation metrics https://semagrow.github.io/kobe/extend/support_metrics

that this one container communicates with the rest (the data sources). By exploiting these premises, KOBE goes further than HOBBIT could have gone to automate the deployment of the modules of an experiment and the control of their connectivity. In other words, KOBE aims at the federated query processing niche and trades off generality of purpose for increased support for its particular purpose.

A similar conclusion is also reached when comparing KOBE with scientific workflow orchestrators. Although (unlike HOBBIT and like KOBE) scientific workflow orchestrators are designed to orchestrate complex systems of containers, they focus on the results of the processing rather on benchmarking the processors. As such, they lack features such as controlling network latency.

Finally, another unique KOBE feature is the mechanism described in Section 4.1 for separating the logs of the different runs of an experiment. This especially useful in stress-testing scenarios where the same query is executed multiple times, so that the query string alone would not be sufficient to separate log lines of the different runs.

## 7 Conclusions

We have presented the architecture and implementation of the KOBE open benchmarking engine for federation systems. KOBE is both open-source software and an open architecture, leveraging containerization to allow the future inclusion of any federation engine. KOBE also uses Elasticsearch as a log server and Kibana as the visualization layer for presenting evaluation metrics extracted from these logs, again emphasizing openness by supporting user-defined ingestion patterns to allow flexibility in how evaluation metrics are to be extracted from each federator's log format. Deployment depends on Kubernetes, which is ubiquitous among the currently prevalent Cloud infrastructures. These features allow experiment publishers the flexibility needed for sharing federated query processing experiments that can be consistently reproduced with minimal effort by the experiment consumers.

Although originally developed for our own experiments, we feel that the federated querying community can extract great value from the abstractions it offers, as it allows releasing a benchmark as a complete, fully configured, automatically deployable testing environment.

As a next step, we are planning to expand the library of federators bundled with the KOBE distribution, and especially with systems that will verify that KOBE operates at the appropriate level of abstraction away from the specifics of particular federators. For instance, adding Triple Pattern Fragments [13] will verify that adaptive source selection and planning can operate within the KOBE framework.

Another interesting future extension would be support for the detailed evaluation of systems that stream results before the complete result set has been obtained. This requires adding support for calculating the relevant metrics, such as the dieﬃciency metric [1].

## Acknowledgments

This project has received funding from the European Union's Horizon 2020 research and innovation programme under grant agreement No 825258. Please see `http://earthanalytics.eu` for more details.

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
