# OpenReview forum: "KOBE: Cloud-native Open Benchmarking Engine for Federated Query Processors"
_eswc-conferences.org/ESWC/2021/Conference/Resources_Track — ESWC 2021 Resources_

### Official Review · AnonReviewer1 · 2021-01-05
**tremendously useful tool suite**

**Rating:** 3
**Confidence:** 5

**Review:**

This resource-track paper introduces KOBE, which is a tool suite that automates the tedious mechanical tasks one has to do when benchmarking query federation engines. That is, after the necessary configuration, KOBE takes care of deploying all relevant components of the experimental setup in a compute cluster, running the experiment, collecting the measurements, and even visualizing the measurements.

As a researcher who is familiar with performing experiments of the types as supported by KOBE, I am very happy to learn about this tool. It sounds like a tremendously useful resource and I will certainly look into using it for future projects.

The paper itself is very well written. It provides a clear motivation, a description of well-justified requirements, and overview of KOBE (its various features and its architecture) that is at the right level of detail. Additionally, the authors also provide a textual comparison to related tools (which are cumbersome to use and require a lot of manual effort, as I can confirm from my own experience).

As high as the quality of the paper is the quality of the online documentation of KOBE. That is, the online documentation is also well written and easy to understand; it provides many examples and goes in detail into the various aspects of using KOBE (including the possible ways to extend it).

Finally, I have also taken a quick look into the source code. It looks well organized and easy to read.

**Anonymity:**

No, I would like my review to be deanonymized.

**Strong Points:**

S1: Extremely useful tool that can save researchers in the community a lot of time.

S2: The tool can even be used to facilitate reproducibility of experiments.

S3: The paper about the resource is well written. It was a pleasure to read.

S4: The corresponding online material is excellent.

**Subreviewer:**

I submitted this review.

**Weak Points:**

None that I can think of.

---

### Official Review · AnonReviewer3 · 2021-01-13
**A valuable tool to regularize the testing of various components involved in federated SPARQL evaluation**

**Rating:** 3
**Confidence:** 4

**Review:**

This submission presents KOBE, a "benchmarking engine" for the evaluation of federated SPARQL systems. The system is comprised of a set of modular components representing dataset and query definitions, SPARQL endpoints, a query federator, and experiment definition. Each of these can be defined independently, and combined so as to allow evaluating the impact of changes to any component (whether that is, e.g., a query federator with improved planning algorithms, or a set of different benchmark queries).

This work is a very good fit for the resources track, providing a tool that allows for the systematic evaluation of federated SPARQL systems. Its use of declarative specifications for experiments and benchmark data and queries enables reproducible evaluations, and its use of container tools such as Docker reduce the burden on those wanting to test new systems and new benchmarks.

As described, this work seems to presume that the federation systems being tested follow a simple setup with a single federator with compartmentalized remote endpoints (a hub and spoke model). This is certainly a reasonable presumption given the existence of many such SPARQL federators, but it should be stated explicitly. If this is in fact the case, could KOBE be extended to support more complex federators (e.g. partial query shipping or endpoints that have mirrors)? Section 3 mentions the ability to define "different networking topologies," which would seem to indicate that more complex cases might be possible. However, there's no elaboration regarding what impact different topologies might have, or what features they might support.

Minor comments:

Some of the links in footnotes (at least those which include an underscore) don't work.

The data visualization choices are a bit strange. For example, the use of a line chart in Figure 3 presents the runtimes of different queries as a continuously connected value. Different presentation style might be more appropriate.


**Anonymity:**

No, I would like my review to be deanonymized.

**Strong Points:**

I appreciate the use of declarative definitions of the various components involved in a benchmarking experiment. I agree with the authors that a prime benefit of this design, including the separation of definition by component, is that it allows those responsible for various components to produce a definition file for the component without concern for unrelated concerns. A benchmark creator can distribute an authoritative definition of the benchmark dataset and queries, and a SPARQL endpoint implementor can distribute a definition indicating the proper configuration of the endpoint. Such separation of concerns would be an obvious benefit to anyone wanting to evaluate a unique configuration of benchmarks, federators, and endpoints.

Regarding the GitHub repository linked in this submission, it appears that the authors have taken time to ensure that the KOBE system is accessible to outside parties. A significant amount of documentation is provided for the described file formats, tooling, and installation guide. Moreover, the authors include a range of example definition files for all the component types, including both those seemingly created/adapted by the authors and those already existing and in use in the community.


**Subreviewer:**

I submitted this review.

**Weak Points:**

It is not entirely clear to me why KOBE is defined in terms of SPARQL federation. While that is certainly a valuable contribution, it seems that KOBE would be just as much a benefit to developers of non-federated systems, or hybrid systems involving a combination of federated and locally-produced results. I think the paper would benefit from a discussion of whether this is an area where KOBE can function and if there are any challenges to serving both federated and non-federated use cases.

Section 4.2 states that "values in each visualization can be also exported in a CSV file for further processing." This is nice, but seems very limiting. I believe a system like KOBE would benefit greatly from being able to generate structured RDF output that captured the specific details of the experiment (benchmark details, software versions used, network topology and configuration, result metrics, etc.). Such output would allow KOBE to be not just a single-use tool to visualize a benchmark evaluation, but a piece of a larger workflow supporting further storage, processing, querying, and analysis of the benchmark experiments.

The discussion of benchmark descriptions in section 5.1 (and all the examples included in the GitHub repository) seems to include only raw SPARQL query strings, and it is not clear if KOBE can support benchmarks such as BSBM which have templated queries which are populated at runtime.

---

### Official Review · AnonReviewer4 · 2021-01-14
**An interesting contribution that needs a little more formality to be applicable community-wide**

**Rating:** 2
**Confidence:** 3

**Review:**

The paper describes a system, called KOBE, which can be seen as a container to accommodate new and existing federated query benchmarks with a focus on the systematic replicability of experiments, orchestration to abate the overheads of network factors, and organization of collected logs to assemble results.

It is important, at this stage, to look back at query federation with awareness of the potential that cloud-native and containerization technologies offer for the benefit of running repeatable benchmarks, so I believe this contribution comes with proper timing. FedBench belongs to an era where the requirements were different, and this work offers an insight as to how it can be rethought-of with more recent requirements. As the authors pointed out, HOBBIT took a step in this direction, which is why I believe they should place the comparison section earlier and elaborate further on KOBE's improvements (which, apart from making the orchestration more transparent, seem to highlight what more it could do than what more it does).

The authors chose a very descriptive narrative style to highlight the merits of the system, which makes the paper easy to follow and a pleasant read overall. This goes at the expense of a more formal delineation of some aspects of the system that I was expecting as I read through. The architecture is illustrated in detail, but for the sake of a resource paper it is important to also specify what the "handles" given to the experimenter look like. For one thing, I would assume the dataset organization and federator designation is present in some manifest with support for certain capabilities: if so, what does this manifest look like? And again, how is does a configuration of log collection into benchmark results look like?  I know, the answer to that lies in the online documentation, which is indeed quite rich, but small excerpts from it in the paper would help rather than harm.

I would also advise that the Fedbench use case in section 4.2 be expanded to show properly as a run-through example, including how it is configured and a snippet of its output.

Minor corrections:
* 2.1 : "so we require that a benchmarking engine use recipes" (infinitive form, lose the "can")
* 3.3 : "new experiment specification*s* that are submitted to KOBE"
* also in 3.3 : "This include*s* the specific initialization process...", "The second phase start*s*..."
* 4.1 : "several experiments can result *in* multiple query evaluations"
* 4.2 : "This visualization can help us..." and "... the experiment execution they belong *to*"




**Anonymity:**

Yes, I would like my review to remain anonymous.

**Strong Points:**

* A welcome contribution as federated query benchmarking does not receive the attention it deserves.
* Good intuition to move towards automation and containerization in the field.
* The writing style makes the paper easy to read.

**Subreviewer:**

I submitted this review.

**Weak Points:**

* A more formal specification of the way federation engines should be described for use in KOBE should be in the paper.
* The FedBench use cases present should be promoted to a run-through example of the capabilities of KOBE.
* The improvements upon HOBBIT seem to be more potential than actual.

---

### Official Review · AnonReviewer2 · 2021-01-17
**Review for KOBE: Cloud-native Open Benchmarking Engine for Federated Query Processors**

**Rating:** 1
**Confidence:** 5

**Review:**


In this paper the authors present a benchmark for SPARQL federation systems that includes a layer for capturing and evaluating the problems these systems may find when deploying over the cloud. This benchmark takes into account the infrastructure by using Kubernetes. The benchmark describes the system to deploy into a configuration file.

The introduction section describes the problem, i.e. current benchmarks do not take into account the infrastructure needed for deploying a benchmark (I agree) nor the network conditions (I disagree). I disagree with the latter since there has been quite a lot of work in adaptive query processing which covers exactly that point.

In Section 2 the authors describe the requirements for the benchmark, including some definitions, data sources and log collection for metrics. I would recommend the authors to look at [1], in which there is a thorough list of requirements for benchmarking federated systems.

In Section 3 the authors present their system, being its main components the deployment subsystem, the networking subsystem and the logging subsystem. For describing the experiments the authors provide a YAML file, which describes the configuration that will be deployed by Kubernetes.

In Section 4 the authors describe how KOBE gets the data from the evaluation and presents it to the user. The question I have here is who is generating the log files that fed the system, the SPARQL federation systems that are evaluated, Kubernetes?

In Section 5 the authors describe that KOBE can be extended, however I missed how can be extended. Maybe with more metrics?

In Section 6 the authors compare with other systems, however I do not see any comparison to adaptive query processing systems. I believe that this is important, since there has been a fair amount of work in that area by database researchers.

In general a nice resource which takes into account the infrastructure dimension of benchmarking a federated system.


[1] Gabriela Montoya, Maria-Esther Vidal, Óscar Corcho, Edna Ruckhaus, Carlos Buil Aranda:
Benchmarking Federated SPARQL Query Engines: Are Existing Testbeds Enough? International Semantic Web Conference (2) 2012: 313-324

**Anonymity:**

Yes, I would like my review to remain anonymous.

**Strong Points:**

A resource that takes into account the infrastructure for evaluating a data federation system

**Subreviewer:**

I submitted this review.

**Weak Points:**

The authors did not take into account research in the adaptive query processing area

---

### Decision · Program_Chairs · 2021-02-23

**Decision:**

Accept

**Comment:**

The resource is a "benchmarking engine" for the evaluation of federated SPARQL systems. The resource eases the tedious task of executing benchmarks and collecting metrics for federated query engines. The paper is easy to read and the resource useful for the semantic web community. Some minor issues have been pointed out in the reviews and have to be considered for the final version.